# A Multicenter Machine Learning-Based Predictive Model of Acute Toxicity in Prostate Cancer Patients Undergoing Salvage Radiotherapy (ICAROS Study)

**DOI:** 10.3390/cancers17132142

**Published:** 2025-06-25

**Authors:** Francesco Deodato, Gabriella Macchia, Patrick Duhanxhiu, Filippo Mammini, Letizia Cavallini, Maria Ntreta, Arina Alexandra Zamfir, Milly Buwenge, Francesco Cellini, Selena Ciabatti, Lorenzo Bianchi, Riccardo Schiavina, Eugenio Brunocilla, Elisa D’Angelo, Alessio Giuseppe Morganti, Savino Cilla

**Affiliations:** 1Radiotherapy Unit, Responsible Research Hospital, 86100 Campobasso, Italy; francesco.deodato@unicatt.it (F.D.); gabriella.macchia@responsible.hospital (G.M.); 2Istituto di Radiologia, Università Cattolica del Sacro Cuore, 00168 Roma, Italy; 3Radiation Oncology, Department of Medical and Surgical Sciences (DIMEC), Alma Mater Studiorum-Bologna University, 40138 Bologna, Italy; patrick.duhanxhiu@studio.unibo.it (P.D.); filippo.mammini@studio.unibo.it (F.M.); milly.buwenge2@unibo.it (M.B.); alessio.morganti2@unibo.it (A.G.M.); 4Radiation Oncology Unit, Radiotherapy Department, Santa Maria della Misericordia Hospital, ULSS5, 45100 Rovigo, Italy; letizia.cavallini@aulss5.veneto.it; 5Radiation Oncology, IRCCS Azienda Ospedaliero-Universitaria di Bologna, 40138 Bologna, Italy; maria.ntreta@aosp.bo.it; 6Dipartimento Universitario Diagnostica per Immagini, Radioterapia Oncologica ed Ematologia, Università Cattolica del Sacro Cuore, 00168 Rome, Italy; francesco.cellini@policlinicogemelli.it; 7Dipartimento di Diagnostica per Immagini, Radioterapia Oncologica ed Ematologia, Fondazione Policlinico Universitario “A. Gemelli” IRCCS, 00168 Rome, Italy; 8Radiation Oncology Department, Bellaria Hospital, AUSL of Bologna, 40139 Bologna, Italy; selena.ciabatti@ausl.bologna.it (S.C.); elisa.dangelo@ausl.bologna.it (E.D.); 9Division of Urology, IRCCS Azienza Ospedaliero-Universitaria di Bologna, 40138 Bologna, Italy; lorenzo.bianchi13@unibo.it (L.B.); eugenio.brunocilla@unibo.it (E.B.); 10Department of Medical and Sciences (DIMEC), Alma Mater Studiorum, Università degli Studi di Bologna, 40138 Bologna, Italy; 11Medical Physics Unit, Responsible Research Hospital, 86100 Campobasso, Italy; savino.cilla@responsible.hospital.it

**Keywords:** acute toxicity, androgen-deprivation therapy, gastrointestinal toxicity, genitourinary toxicity, LASSO regression, machine learning, planning target volume, predictive model, prostate cancer, salvage radiotherapy

## Abstract

Salvage radiotherapy is offered to men whose prostate-specific antigen starts to rise again after surgery for prostate cancer, giving them a second chance of cure. Most patients tolerate the treatment well, but some experience bothersome bladder or bowel symptoms during the first three months. Using real-world data from 454 Italian patients and an intuitive machine-learning approach, we found that three factors mainly influence these short-term side-effects: the surgical technique used to remove the prostate (traditional open surgery versus minimally invasive laparoscopic or robotic approaches), the safety margin added around the target in the radiotherapy plan, and how extensively pelvic lymph nodes were removed. Understanding these factors can help doctors design gentler treatment plans and monitor higher-risk patients more closely, allowing them to start simple supportive measures early and better preserve quality of life.

## 1. Introduction

Prostate cancer remains a significant global health concern, with incidence and mortality rates influenced by geographical, socio-economic, and healthcare disparities. As its burden increases, proactive strategies are essential to improve patient outcomes and alleviate healthcare system strain [1]. Treatment options for localized prostate cancer include active surveillance, radical prostatectomy (RP), and radiotherapy, while advanced cases require androgen-deprivation therapy (ADT), chemotherapy, and novel androgen signaling inhibitors. RP remains a cornerstone of curative management for localized disease [2].

Despite its effectiveness, approximately 25% of patients experience biochemical recurrence within ten years post-RP, necessitating effective salvage therapies [3]. Salvage radiotherapy (SRT) has become the standard treatment for biochemical recurrence, offering potential disease control and improved survival outcomes [2]. However, SRT is associated with both acute and late toxicities, significantly impacting patients’ quality of life [4]. Identifying predictive factors for toxicity is crucial for optimizing treatment, enabling personalized dosimetry adjustments, improving patient selection, and setting realistic expectations for patients undergoing SRT.

Late toxicity in SRT has been widely studied due to its long-term impact on quality of life, but acute toxicity remains clinically relevant, as it can compromise treatment adherence, reduce patient comfort, and contribute to consequential late toxicity [5]. Despite its significance, limited research has focused on predictive factors for acute toxicity. Few studies, such as those by Ost et al. [6] and Ghadjar et al. [7], have specifically addressed acute toxicity, while many investigations have examined late toxicity [8] or included patients treated with postoperative/adjuvant radiotherapy [9,10,11,12] or mixed cohorts receiving adjuvant [13,14] or exclusive radiotherapy [15,16].

Existing studies on SRT have primarily investigated isolated factors such as radiation dose [7] or image-guidance techniques [6], rather than conducting a comprehensive analysis of patient- and treatment-related predictors of acute toxicity. Given the complexity of toxicity prediction in post-prostatectomy radiotherapy, the traditional statistical methods have limitations. Advanced variable selection techniques, such as Least Absolute Shrinkage and Selection Operator (LASSO) regression, and AI-based approaches using different machine learning (ML) methods may offer significant advantages. These methods enhance model accuracy, mitigate overfitting, and facilitate personalized predictions by integrating multiple prognostic factors, including clinical and tumor characteristics, imaging indices, and anthropometric data.

Based on this background, we conducted a comprehensive analysis using LASSO and different ML techniques to refine the identification of key predictive factors for acute toxicity during SRT for prostate cancer. By leveraging these advanced techniques for variable selection and modeling, our study aims to advance current predictive models, improving clinical decision-making and patient outcomes in this setting.

## 2. Materials and Methods

### 2.1. Study Design and Endpoints

This study is a retrospective analysis of patients enrolled in a multicentric observational study. The aim was to develop a predictive model for acute GI and GU toxicity. Acute grade ≥ 2 GU and GI toxicities (RTOG scale) recorded from the start of radiotherapy to 90 days post-treatment were chosen as the clinical endpoints because grade 2 represents the first toxicity level at which symptoms (e.g., dysuria requiring α-blockers, ≥4 extra bowel movements per day needing loperamide) interfere with daily activities and necessitate medical intervention, thereby impacting quality of life and potentially prompting treatment adaptation. In particular, both toxicity responses were treated as a binary indicator (grade < 2 vs. grade ≥ 2).

Treatment characteristics in this cohort, including dose, fractionation, prophylactic lymph-node irradiation and combination with ADT, varied according to departmental guidelines and the judgment of the attending physicians. Treatment protocols were not uniform across participating centers; they evolved over time in line with available technological resources (e.g., transition from three-dimensional conformal radiotherapy [3D-CRT] to intensity-modulated radiotherapy/volumetric modulated arc therapy [IMRT-VMAT]), the progressive spread of mild hypofractionation in prostate cancer, and the personal preferences of the responsible radiation oncologists. Setup verification was performed using electronic portal imaging devices (EPIDs) until 2015 and cone beam computed tomography (CBCT) thereafter, as previously described [17].

### 2.2. Inclusion and Exclusion Criteria

Eligible patients were men with histologically confirmed prostatic adenocarcinoma treated by RP who showed no evidence of distant metastases on conventional imaging or PSMA-PET and had biochemical recurrence, defined as at least two consecutive PSA rises with the last value greater than 0.2 ng/mL; all patients received external-beam photon salvage radiotherapy (3D-CRT, IMRT, or VMAT), and the presence of macroscopic local recurrence in the prostate bed did not preclude inclusion. Patients were excluded only if they had previously undergone pelvic radiotherapy, had absolute contraindications to pelvic irradiation such as active inflammatory bowel disease, pelvic abscess, and fistula, or had already received androgen-deprivation therapy as salvage treatment for biochemical recurrence. No upper age limit was applied, and a history of other malignancies, TURP, benign prostatic hyperplasia, or hormonal therapy unrelated to prostate cancer did not influence eligibility.

### 2.3. Evaluated Parameters

The parameters analyzed in this study included patient demographic and clinical characteristics, treatment details, and radiotherapy-related variables. Patient-related parameters included age and the age-adjusted Charlson Comorbidity Index (CCI). Surgical factors considered were the technique used (open, laparoscopic, or robotic surgery) and whether lymphadenectomy was performed and, if so, the extent of dissection, classified as limited (<15 nodes removed) or extended (≥15 nodes removed); the cut-off of 15 nodes was adopted because an extended PLND with a median of 15 nodes excised is associated with only a moderate increase in surgical morbidity compared with limited dissection and lower complication rates than the super-extended approach that typically removes twenty or more nodes [18]. Treatment-related parameters included the use of prophylactic nodal irradiation, presence of macroscopic recurrence in the prostate bed, and a history of previous abdominal–pelvic surgery. ADT was analyzed in terms of its administration and the specific type of therapy used, such as luteinizing hormone-releasing hormone (LH-RH) agonist or high-dose bicalutamide. Radiotherapy-specific variables included the fractionation regimen (hypofractionated, with dose per fraction > 2 Gy, or conventional), the technique employed (3D-CRT or IMRT-VMAT), and the image guidance modality (EPID or CBCT). Additionally, dosimetric parameters such as the equivalent dose in 2 Gy fractions (EQD2) for the prostate and lymph nodes were evaluated. Acute toxicity was assessed during weekly on-treatment visits and at the first follow-up visit, conducted three months after the completion of radiotherapy. Acute toxicity was evaluated using the Radiation Therapy Oncology Group (RTOG) scale [19] and it was defined as any toxicity recorded from the start of radiotherapy to 90 days post-treatment.

### 2.4. Flowchart and Data Preparation

Figure 1 reports an overview of the flowchart of this study in four steps. In the first step, patient demographic, clinical characteristics, treatment details, and radiotherapy-related variables were collected and analyzed from a total of 454 patients treated across three Italian radiotherapy departments. Eventual missing clinical data were assumed missing at random and nondependent on outcome. All variables with a missing percentage < 20% were retained and filled in with the missForest method. Multicategory variables were processed using one-hot encoding. In the second step, we performed a univariate analysis followed by the least absolute shrinkage and selection operator regression method (LASSO) in order to identify potential variables associated with GI and GU acute toxicities, to avoid overfitting and reduce the dimensionality of variables. LASSO regression was performed using the “glmnet” package in the R software package (Version 4.4.3). Chi-square and Fisher’s exact tests were applied to categorical variables; independent *t*-tests and Mann-Whitney U-tests were used for continuous variables. Thus, the dataset was partitioned into training and validation sets in a 70:30 ratio with a balanced toxicity incidence rate. In the following step, the selected variables were used in the training and validation of five different classification machine learning models to predict acute toxicity, as explained with greater details in Section 2.5. All models were evaluated in terms of discrimination and calibration. Lastly, in the final step, two predictive models based on decision trees were developed for GI and GU toxicities.

### 2.5. Machine Learning Modeling and Statistical Analysis

Key prognostic variables were identified through the LASSO regression model by selecting only the variables with nonzero coefficients. These selected variables were then utilized to construct five different classification machine learning models to predict acute toxicity, including logistic regression (LR), Gaussian Naïve Bayes (GNB), K-means neighbor (KNN), decision tree (DT), and Light Gradient-Boosting Machine (LGBM).

The dataset was partitioned into training and validation sets in a 70:30 ratio with a balanced toxicity incidence rate, in order to ensure that similar proportions of the outcomes were preserved in each set. The Z-score was used to standardize variables with varying dimensionality. The training cohort was utilized to generate the prediction models and the remaining 30% was employed to estimate the models’ performance. To maximize the prediction performance, hyperparameters of all classifiers were tuned using a randomized search with 100 iterations, implemented using the RandomizedSearchCV algorithm in Python. The effectiveness of each model was evaluated in terms of discrimination and calibration. Model performance was evaluated using accuracy, F1-score, and areas under their receiver operating characteristic curve (AUC) metrics. The uncertainty of model’s performance was quantified by bootstrap analysis, with 95% confidence intervals (CI) intervals based on 100 bootstrap samples. Specifically, for each resample, a new training set was generated by randomly selecting patients with replacements from the original training dataset. The remaining patients were used as validation cohorts. This process was repeated 100 times to assess the robustness of the models. Calibration was performed applying the isotonic regression method, and it was evaluated using the calibration plots and the Brier score metric (i.e., the squared differences between predicted and observed outcomes, estimated for each model). This score ranges between 0 and 1, and lower values indicate better performance.

Continuous data are reported with median and range, the categorical data are reported with frequencies and percentage. The Fishers exact test for categorical variables and the Wilcoxon rank sum test for continuous variables are used to obtain *p*-values for statistical significance. The XLSTAT statistical software package (Version 2019.2.2) (Addinsoft, New York, NY, USA) and Python 3.8 (Python Software Foundation, Beaverton, OR, USA) were used for statistical analysis.

### 2.6. Ethical Issues

The local institutional review board approved this analysis (311/2019/Oss/AOUBo, ICAROS-1 study). Only patients who had provided written informed consent to the scientific use of their data were included. All the study procedures were compliant with the 2021 WHO guidance on ethics and governance of artificial intelligence for health and with the TRIPOD+AI statement [20,21].

## 3. Results

### 3.1. Patients’ Characteristics

This analysis included a total of 454 patients treated across three Italian radiotherapy departments. Overall, 64% of patients received androgen-deprivation therapy (ADT), of whom 47.4% were treated with a luteinizing hormone-releasing hormone (LH-RH) agonist (triptorelin acetate 11.25 mg every three months or leuprorelin acetate 11.25 mg every three months) or with high-dose bicalutamide (150 mg daily). Patients, tumor, and treatment characteristics, including pathological TNM stage, Gleason score, D’Amico risk classification, and the full radiotherapy fractionation scheme, are detailed in Table 1. Concomitant medical therapies other than androgen-deprivation therapy were not collected in the source databases and are therefore unavailable for analysis. A total of 318 (70%) patients were randomized to the training cohort and 136 patients (30%) to the validation cohort. In the training cohort, 90 and 68 patients were associated with ≥2 GI and ≥2 GU toxicities, respectively. Similarly, in the validation cohort, 38 and 31 patients were associated with ≥2 GI and ≥2 GU toxicities, respectively.

As reported in Table 1, the two cohorts were closely matched for most of collected variables.

### 3.2. Acute Toxicity

The analysis of acute toxicity revealed a balanced distribution of GI and GU toxicities among patients, with no cases of severe toxicity (grade ≥ 4) observed. For GI toxicity, 326 patients (71.8%) experienced no symptoms (grade 0) or reported mild toxicity (grade 1). Moderate toxicity (grade 2) was observed in 124 patients (27.3%), and only 4 patients (0.9%) experienced more severe symptoms (grade 3).

Similarly, for GU toxicity, 355 patients (78.2%) had no reported toxicity (grade 0) or experienced mild symptoms (grade 1). Moderate toxicity (grade 2) occurred in 96 patients (21.1%), with only 3 patients (0.7%) experiencing grade 3 symptoms.

### 3.3. Variables Selection

After a univariate, LASSO regression and a collinearity analysis, all used to screen the independent related factors of acute toxicities, the number of variables associated with acute toxicities was reduced to eight for both groups. These variables included the type of surgery (Surg-Tech), the CTV-to-PTV margin (CTV-PTV), the biological equivalent total dose to prostatic fossa (EQD2), the dose per fraction (FraD), the age-adjusted Charlson Comorbidity Index (CCI), lymphadenectomy (LA), the radiotherapy technique (RT-tech), the type of hormonal therapy (HT), and the use of IGRT. The optimal λ values, as determined by cross validation was 0.0171 and 0.0155 for GI and GU acute toxicities, respectively. Figure 2a,c show the binomial deviation as a function of the tuning penalization parameter λ. The optimal λ value corresponds to the minimum of this function. Figure 2b,d show the heatmap plots representing the correlation matrixes between variables after LASSO selection.

### 3.4. Machine Learning Models

The predictive performances of the different classifiers in the validation cohort are reported in Table 2. The ROC curves for the five ML models in the training and validation cohorts for GI and GU acute toxicities are shown in Figure 3.

The best AUC scores for both GI and GU endpoints were found for the DT and LGBM models. For GI toxicity, the DT and LGBM reported AUC values of 0.776 (95% CI: 0.715–0.832) and 0.784 (95% CI: 0.725–0.843) in the training cohort and 0.690 (95% CI: 0.655–0.79) and 0.706 (95% CI 0.664–0.747), in the validation cohorts, respectively. Similarly, for GU toxicity, the DT and LGBM reported AUC values of 0.905 (95% CI: 0.840–0.975) and 0.911 (95% CI: 0.843–0.979) in the training cohort, and 0.848 (95% CI: 0.809–0.891) and 0.855 (95% CI: 0.814–0.896) in the validation cohorts, respectively.

The calibration curves for the validation set of the different models reported a good agreement between the predicted and observed values, with DT and LGBM showing the best performances (Figure 4). Brier scores ranged from 0.09 for the DT and LGBM models to 0.21 for the KNN model.

Therefore, we ultimately chose the DT model to obtain a hierarchical flowchart-like structure, providing a clear and understandable path of decisions.

### 3.5. Predictive Model of Gastrointestinal Toxicity

The classification tree for the most informative variables is displayed in Figure 5.

Node 1 illustrates the distribution of initial GI toxicity, with 28.2% of patients having grade ≥ 2 GI toxicity. Nodes 2 and 3 show the resulting classification after the adding of the strongest predictive variable. The primary determinant of acute GI toxicity was the surgical technique. Patients who underwent open surgery exhibited the highest toxicity rate, with 41.8% experiencing grade ≥ 2 toxicity. Within this group, the clinical target volume (CTV) to planning target volume (PTV) margin emerged as a significant factor. Patients with a margin of less than 10 mm had a toxicity rate of 36.9%, while those with a margin of 10 mm or more experienced a markedly higher rate of 70.4%. For the subgroup with a CTV-to-PTV margin of less than 10 mm, the extent of lymphadenectomy provided further stratification. Those with fewer than 15 nodes resected had a toxicity rate of 31.8%, whereas patients with 15 or more nodes resected showed a significantly increased rate of 48.9%. No further stratification was evident for patients with a CTV-to-PTV margin of 10 mm or greater.

In contrast, patients who underwent laparoscopic or robotic surgery had a much lower overall toxicity rate of 18.9%. Among these patients, the EQD2 was the key factor influencing toxicity. For those who received an EQD2 of less than 70 Gy, the toxicity rate was 12.9%. Within this group, ADT further differentiated risk. Patients treated with an LH-RH agonist had a higher toxicity rate of 18.9%, while those who received no ADT or high-dose bicalutamide (HDB) experienced a much lower rate of 8.3%. For patients receiving an EQD2 of 70 Gy or more, the toxicity rate was slightly higher at 21.6%. In this subgroup, the CCI was the next relevant factor. Patients with a CCI score of less than 3 showed a toxicity rate of 17.9%, whereas those with a score of 3 or greater had an elevated rate of 25.6%.

### 3.6. Predictive Model of Genitourinary Toxicity

The classification tree for the most informative variables is displayed in Figure 6.

The primary factor influencing acute GU toxicity was the surgical technique. Patients who underwent open surgery had a significantly higher toxicity rate of 35.9%, compared to those who underwent laparoscopic or robotic surgery, where the rate was much lower at 12.2%. For the latter, no further stratification was recorded. Among patients in the open surgery group, the CTV-to-PTV margin further stratified toxicity risk. Those with a margin of less than 10 mm exhibited a toxicity rate of 29.9%, while those with a margin of 10 mm or greater had a markedly higher rate of 70.4%. Within the subgroup of patients with a CTV-to-PTV margin of less than 10 mm, the extent of lymphadenectomy was an additional determinant. Patients with fewer than 15 nodes resected during lymphadenectomy had a toxicity rate of 22.7%. In contrast, those with 15 or more nodes resected experienced a significantly increased rate of 46.8%. No further stratification was observed for patients with a CTV-to-PTV margin of 10 mm or greater.

## 4. Discussion

This study aimed to develop a predictive model for acute GI and GU toxicity in prostate cancer patients undergoing SRT. Using data from 454 patients across three centers, we applied different machine learning techniques, to identify key toxicity predictors. Surgical technique, CTV-to-PTV margins, and radiotherapy parameters emerged as the main determinants, with open surgery and larger margins associated with higher grade ≥ 2 acute toxicity.

Among the ML classifiers, LGBM and DT models demonstrated the best performance in terms of model discrimination and calibration. In particular, because of their clinical interpretability, DT models were used to create the two risk models for acute toxicities. In this study we demonstrated that DT is an easy-to-understand ML technique, able to handle non-linear relationships between variables and presenting a tree-structure simple to understand also for non-technical users. This is a relevant topic because rather than relying on complex black-box algorithms, clinicians may understand the role and impact of the different predictive variables on the risk of toxicity, and assume more informed decisions.

Our findings confirm the excellent tolerability of SRT, with no patients experiencing grade ≥ 4 toxicity and grade 3 toxicity remaining below 1%. Importantly, the manageable nature of grade 2 events means that toxicity risk should not deter offering SRT to suitable patients with biochemical recurrence; instead, our model allows clinicians to identify individuals at higher risk, monitor them more closely during treatment and the early post-treatment period, and initiate supportive therapies (such as α-blockers or loperamide) at the first sign of symptoms. The DT analysis shows how minimally invasive prostatectomy (laparoscopic or robotic) was linked to lower toxicity rates, aligning with its well-documented advantages, including reduced blood loss, shorter recovery, and improved functional outcomes [2]. The extent of lymphadenectomy further influenced GI toxicity, particularly in open surgery cases.

The impact of surgical technique on radiation-induced toxicity may be due to greater tissue manipulation and inflammatory response associated with open prostatectomy. While this approach is often extraperitoneal [22], laparoscopic and robotic prostatectomy are typically transperitoneal [23], which may reduce postoperative inflammation. The more extensive tissue disruption in open surgery may heighten pelvic tissue sensitivity to radiation, exacerbating acute toxicity. Furthermore, laparoscopic and robotic prostatectomy are associated with better lymphatic drainage, reducing the risk of complications such as lymphedema and lymphocele [24,25]. These findings reinforce the benefits of minimally invasive surgery for optimizing both oncological and functional outcomes.

Our results align with prior studies on post-prostatectomy radiotherapy (Table 2). Ghadjar et al. [7] found no significant increase in acute GU toxicity with higher radiation doses, a trend observed in our cohort. Similarly, Buwenge et al. [11] demonstrated the safety of hypofractionation, which we confirm. However, some discrepancies exist. Ost et al. [6] reported reduced acute toxicity with CBCT-based image guidance, whereas we found no significant difference, likely due to the limited proportion of CBCT-treated patients in our cohort. Additionally, while some studies [9,14] reported increased acute toxicity with prophylactic pelvic nodal irradiation, our data did not show significant differences, possibly due to variations in patient populations, treatment protocols, and radiation dose distribution.

Recently, a number of studies have employed ML-based models for the prediction of radiotherapy-related side effects, including acute and late gastrointestinal and genitourinary toxicities in prostate treatments [26,27,28,29,30,31,32,33]. Back in 2011, Pella et al. [27] made a first attempt to apply ML techniques for the prediction of acute toxicity for urinary bladder and rectum, reporting a prediction accuracy of AUC = 0.7 using an artificial neural network derived from a cohort of 321 patients. Ospina et al. [28] proposed a random forest normal tissue complication probability model to predict late rectal toxicity following prostate cancer radiation therapy, and to compare its performance to that of classic NTCP models. The age and use of anticoagulants were found to be predictors of rectal bleeding, with an AUC ranging from 0.66 to 0.76, depending on the toxicity endpoint. The model included variables other than the dosimetric ones, and was considered as a strong competitor to classic NTCP models. Jones et al. [31] sought to help identify patients at higher risk of developing rectal injury based on estimated rectal dosimetry compliance prior to planning procedure. Logistic regression, classification and regression tree, and random forest models were compared for their ability to discriminate between class outcomes. Both logistic regression and random forest modeling approaches demonstrated good discriminative ability for predicting class outcomes in the upper dose levels. More recently, the role of genomics and radiomic features as new variables in machine learning modeling was also evaluated to predict radiation-induced rectal toxicities [31,32]. Lee et al. [32] developed a random forest regression method for predicting the risk of GU toxicity by identifying and integrating patterns in genome-wide single-nucleotide polymorphisms. A gene ontology analysis highlighted key biological processes, such as neurogenesis and ion transport, from the genes known to be important for urinary tract functions. In the study of Yang et al. [33], 183 patients were included and toxicity scores were prospectively collected after 2 years with grade ≥ 1 proctitis, hemorrhage and GI recorded as the endpoints of interest. The test set AUC values were 0.549, 0.741, and 0.669 for proctitis, hemorrhage, and GI toxicity prediction using radiomic combined with dosimetric features.

However, the application of these approaches in clinical routine is still lagging, mainly due to their low interpretability; i.e., they are uninterpretable with respect to the importance of each feature on the output prediction and are generally regarded as “black boxes”. This is undesirable because clinicians will lose trust as a result of the prediction’s unclear understanding.

This study has several strengths. The large sample size and comprehensive evaluation of multiple patient-, surgical-, and radiotherapy-related factors enhance the robustness of our findings. Furthermore, the use of advanced variable selection techniques such as LASSO regression and machine learning algorithms such as decision tree improves prediction accuracy and clinical applicability by efficiently handling complex datasets. These methods help prioritize the most relevant predictors and provide a decision-making framework that is easily interpretable in clinical practice. In particular, decision trees facilitate the early identification of patients at risk and the timely utilization of mitigative measures to tackle the toxicities issues.

However, some limitations must be acknowledged. First, our dataset did not include the interval between prostatectomy and salvage radiotherapy, peri-operative surgical complications, or baseline genitourinary/gastrointestinal symptoms, all of which may influence acute toxicity. Future prospective studies should collect these variables systematically. A further limitation is the relatively low proportion of patients treated with robotic prostatectomy and the limited use of CBCT-based daily IGRT, which is now common practice. However, the multicenter nature of the cohort reflects real-world heterogeneity, and the identified predictors, particularly CTV-to-PTV margin, remain applicable to modern workflows, where careful balance is needed between margin reduction and the risk of geographic miss. Moreover, grade 2 toxicity, while relevant, may be less clinically significant than higher-grade toxicities. Certain parameters, including specific dosimetric constraints [12,15,16] and baseline symptoms [15], were not available in our dataset. Additionally, the 10 mm CTV-to-PTV margin, though considered large by current IGRT standards, was based on 2008 guidelines [34] and remained standard practice in our centers for several years. The retrospective nature of the study and differences in surgical approaches across centers may also have influenced findings. While our results provide valuable insights, prospective validation in larger, independent cohorts is needed to confirm their clinical applicability.

A key clinical recommendation is to reduce the CTV-to-PTV margin to <10 mm, particularly in open prostatectomy cases, to minimize acute toxicity. This can be achieved using IGRT or motion management techniques such as rectal balloons [35]. The integration of AI-driven predictive models into routine clinical practice may further refine treatment personalization and enhance patient outcomes.

Future studies should validate these findings in independent cohorts and incorporate additional parameters, such as medication use, lifestyle factors, and comorbidities, to further enhance predictive models. The role of emerging radiotherapy techniques, such as adaptive radiotherapy, should also be explored in the context of minimizing toxicity while maintaining treatment efficacy. Finally, our database, though extensive, was initially established nearly a decade ago and did not include emerging predictors potentially associated with toxicity, such as inflammatory indices, body-composition metrics (e.g., sarcopenia), or genetic biomarkers. A prospective study is currently being planned at our institution to specifically investigate inflammation indices and body composition in prostate cancer patients, assessing their potential effects on both clinical outcomes and acute and late radiotherapy toxicity.

## 5. Conclusions

This study highlights the interplay between patient characteristics, surgical techniques, and radiotherapy parameters in determining acute toxicity after SRT. Decision tree ML-based models demonstrated the ability to accurately assess the risk of GI and GU acute toxicities, providing an intelligible explanation of individualized risk prediction. Our findings support the importance of personalized treatment strategies to minimize toxicity and optimize patient outcomes.

## Figures and Tables

**Figure 1 cancers-17-02142-f001:**
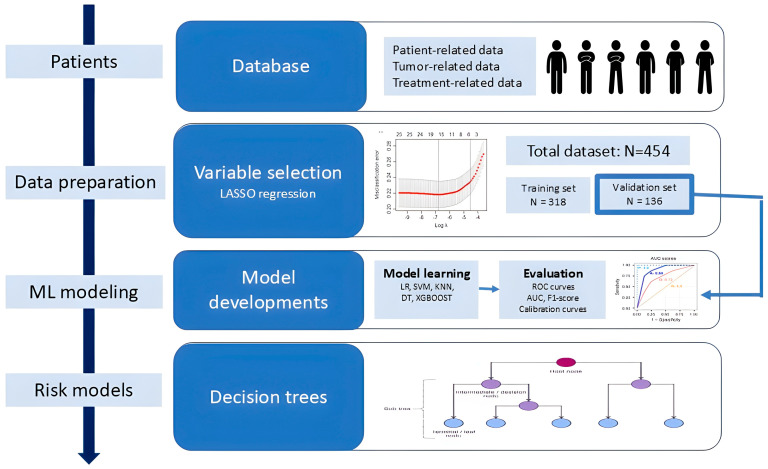
Overall flowchart of the study.

**Figure 2 cancers-17-02142-f002:**
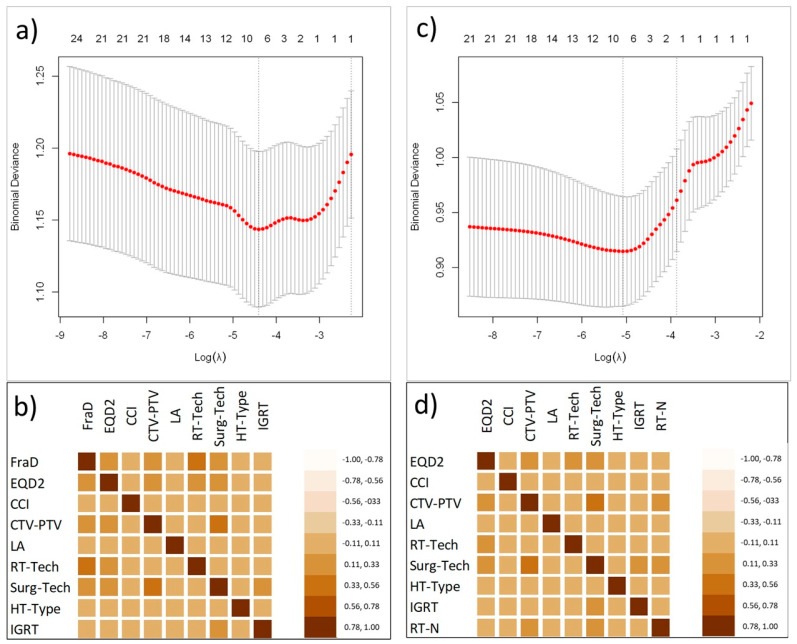
Variable selection using the least absolute shrinkage and selection operator (LASSO) binary logistic regression model. (**a**,**c**) Binomial deviation as a function of the tuning penalization parameter λ. The numbers at the top of the chart are the remaining coefficients at the corresponding log lambda values. (**b**,**d**) Correlation matrix heatmaps.

**Figure 3 cancers-17-02142-f003:**
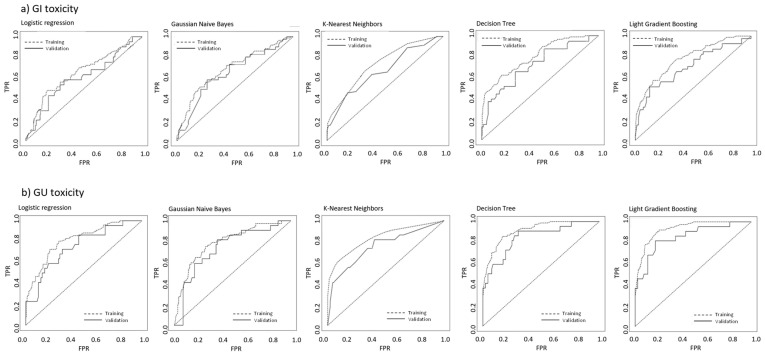
ROC curves for the training and validation cohorts of the five different machine algorithms.

**Figure 4 cancers-17-02142-f004:**
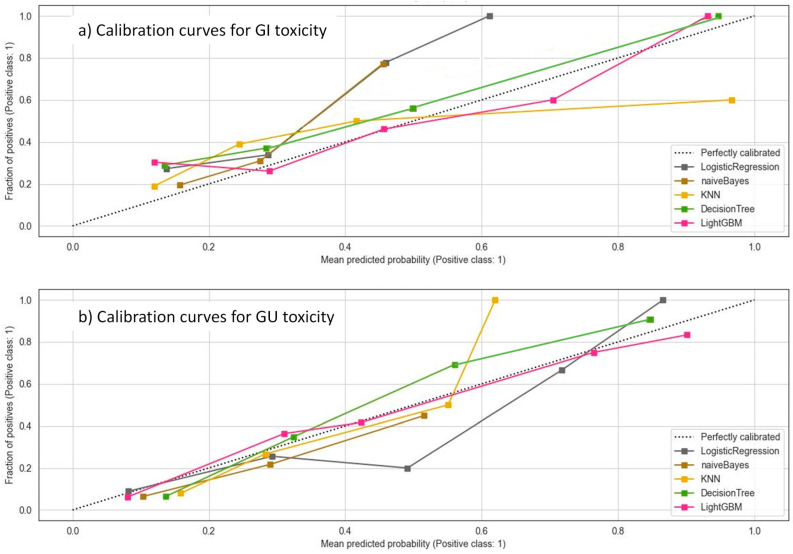
Calibration curves for the five ML models on the validation set. The dotted line represents the perfect calibration curve; i.e., the predicted probability matches the true probability perfectly.

**Figure 5 cancers-17-02142-f005:**
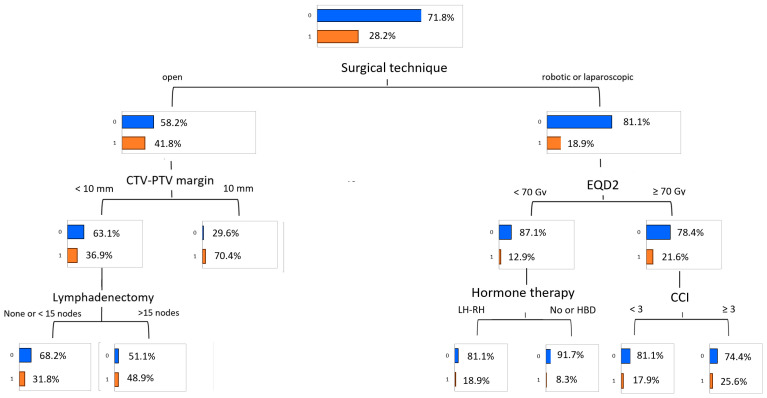
Predictive model of acute gastrointestinal toxicity (grade ≥ 2) obtained by the Decision Tree algorithm. The values highlighted close to the red bars denote the percentages of acute toxicity incidence.

**Figure 6 cancers-17-02142-f006:**
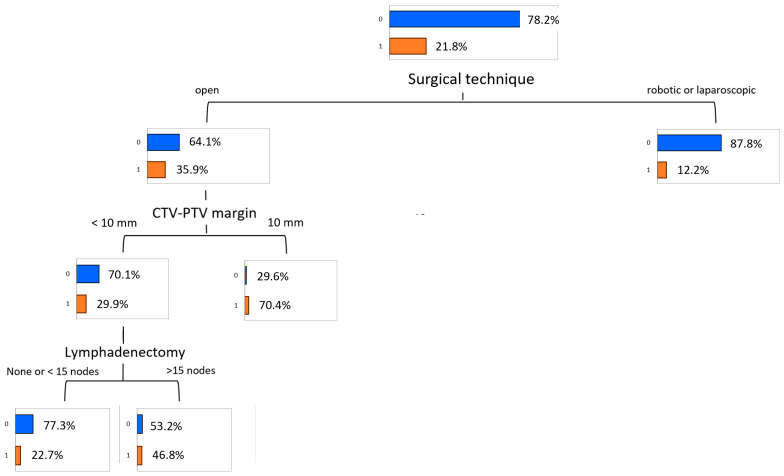
Predictive model of acute genitourinary toxicity (grade ≥ 2) obtained by the Decision Tree algorithm. The values highlighted close to the red bars denote the percentages of acute toxicity incidence.

**Table 1 cancers-17-02142-t001:** Patient and treatment characteristics. Abbreviations 3DCRT: three-dimensional conformal radiotherapy; α/β: alpha–beta ratio; CBCT: cone beam computed tomography; EPID: electronic portal imaging device; EQD2: equivalent dose in 2 Gy fractions; Gy: gray; IMRT: intensity-modulated radiotherapy; LH-RH analog: luteinizing hormone-releasing hormone analog; VMAT: volumetric modulated arc therapy. ISUP: International Society of Urological Pathology; EAU: European Association of Urology.

Variable	Value	Total (N, %)	Training (N, %)	Validation (N, %)	*p*-Value
GI toxicity	<2	326 (71.8)	228 (71.7)	98 (72.1)	0.812
	≥2	128 (28.2)	90 (28.3)	38 (27.9)	
GU toxicity	<2	355 (78.2)	250 (78.6)	105 (77.2)	0.752
	≥2	99 (21.8)	68 (21.4)	31 (22.8)	
Age-adjusted Charlson Comorbidity Index	<3	198 (43.6)	138 (43.4)	60 (44.1)	0.802
	≥3	256 (56.4)	180 (56.6)	76 (55.9)	
ISUP Grade	1	77 (17.0)	57 (17.9)	20 (14.7)	0.114
	2	83 (18.3)	55 (17.3)	28 (20.6)	
	3	117 (25.8)	75 (23.6)	42 (30.9)	
	4	86 (18.9)	68 (21.4)	18 (13.2)	
	5	91 (20.0)	61 (19.2)	30 (22.1)	
pT stage	1	4 (0.9)	3 (0.9)	1 (0.7)	0.414
	2	183 (40.3)	124 (39.0)	59 (43.4)	
	3	261 (57.5)	187 (58.8)	74 (54.4)	
	4	6 (1.3)	4 (1.3)	2 (1.5)	
pN stage	0	392 (86.3)	272 (85.6)	120 (88.2)	0.888
	1	62 (13.7)	46 (14.4)	16 (11.8)	
EAU risk category	Very low/low	11 (2.4)	9 (2.8)	2 (1.5)	0.228
	Intermediate	38 (8.4)	24 (7.5)	14 (10.3)	
	High	263 (57.9)	195 (61.4)	68 (50.0)	
	Locally advanced	142 (31.3)	90 (28.3)	52 (38.2)	
Surgical technique	Open	184 (40.5)	135 (42.5)	49 (36.0)	0.175
	Laparoscopic	216 (47.6)	155 (48.7)	61 (44.9)	
	Robotic	54 (11.9)	28 (8.8)	26 (19.1)	
Nodal irradiation	No	184 (40.5)	138 (43.4)	46 (33.8)	0.151
	Yes	270 (59.5)	180 (56.6)	90 (66.2)	
Lymphadenectomy	No	178 (39.2)	112 (35.2)	66 (48.5)	0.369
	<15 nodes	137 (30.2)	104 (32.7)	33 (24.3)	
	≥15 nodes	139 (30.6)	102 (32.1)	37 (27.2)	
Macroscopic recurrence in the prostate bed	No	371 (81.7)	268 (84.3)	103 (75.7)	0.378
	Yes	83 (18.3)	50 (15.7)	33 (24.3)	
Previous abdominal–pelvic surgery	No	417 (91.9)	299 (94.0)	118 (86.8)	0.275
	Yes	37 (8.1)	19 (6.0)	18 (13.2)	
Androgen-deprivation therapy	No	163 (35.9)	117 (36.8)	46 (33.8)	0.775
	Yes	291 (64.1)	201 (63.2)	90 (66.2)	
Type of androgen-deprivation therapy	Not prescribed	163 (35.9)	125 (39.3)	38 (27.9)	0.188
	LH-RH analog	215 (47.4)	150 (47.2)	65 (47.8)	
	High-dose Bicalutamide	76 (16.7)	43 (13.5)	33 (24.3)	
Radiotherapy technique	3D-CRT	119 (26.2)	74 (23.3)	45 (33.1)	0.942
	IMRT/VMAT	335 (73.8)	244 (76.7)	91 (66.9)	
Image guidance	EPID	367 (80.8)	251 (78.9)	116 (85.3)	0.782
	CBCT	87 (19.2)	67 (21.1)	20 (14.7)	
CTV-to-PTV margin (mm)	Median (range)	10 (6–10)	10 (6–10)	10 (6–10)	0.512
EQD2 to the prostatic fossa α/β_10_ (Gy)	Median (range)	69.0 (60.0–80.0)	69.0 (60.0–80.0)	68.3 (65.1–80.0)	0.612
EQD2 to the lymph nodes α/β_10_ (Gy)	Median (range)	44.3 (44.3–53.1)	44.3 (44.3–53.1)	44.3 (44.3–53.1)	0.950
Total dose to the prostatic fossa (Gy)	<70 Gy	264 (58.1)	190 (59.7)	74 (54.4)	0.973
	≥70 Gy	190 (41.9)	128 (40.3)	62 (45.6)	
Dose per fraction to the prostatic fossa (Gy)	≤2.0 Gy	168 (37.0)	113 (35.5)	55 (40.4)	0.368
	2.1–2.4 Gy	122 (26.9)	90 (28.3)	32 (23.5)	
	>2.4 Gy	164 (36.1)	115 (36.2)	49 (36.0)	

**Table 2 cancers-17-02142-t002:** Predictive performances of the different ML models to predict acute toxicity following postoperative radiotherapy for prostate cancer in the validation cohort.

	Assessment Metrics (95% CI)			
ML Model	ROC AUC	Accuracy	F1-Score	Brier Score
GI toxicity				
Logistic regression	0.632 (0.578–0.686)	0.638 (0.597–0.679)	0.712 (0.666–0.758)	0.194 (0.180–0.208)
Naive Bayes	0.624 (0.572–0.676)	0.599 (0.563–0.635)	0.646 (0.607–0.685)	0.201 (0.186–0.214)
K-nearest neighbors	0.649 (0.597–0.701)	0.633 (0.601–0.665)	0.721 (0.685–0.757)	0.215 (0.201–0.229)
Decision tree	0.690 (0.641–0.739)	0.668 (0.638–0.698)	0.755 (0.721–0.789)	0.123 (0.116–0.131)
LightGBM	0.706 (0.657–0.755)	0.703 (0.675–0.731)	0.766 (0.735–0.797)	0.121 (0.114–0.128)
GU toxicity				
Logistic regression	0.751 (0.699–0.803)	0.722 (0.675–0.769)	0.788 (0.737–0.839)	0.181 (0.169–0.193)
Naive Bayes	0.738 (0.688–0.788)	0.685 (0.644–0.726)	0.752 (0.707–0.797)	0.192 (0.179–0.205)
K-nearest neighbors	0.769 (0.719–0.819)	0.751 (0.713–0.789)	0.816 (0.775–0.857)	0.188 (0.176–0.200)
Decision tree	0.848 (0.805–0.891)	0.784 (0.749–0.819)	0.854 (0.816–0.892)	0.092 (0.087–0.098)
LightGBM	0.855 (0.814–0.896)	0.821 (0.780–0.862)	0.878 (0.843–0.913)	0.088 (0.083–0.093)

## Data Availability

The data of this study are available from the corresponding author upon reasonable request to the corresponding author https://doi.org/10.5281/zenodo.14832521.

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
