# Peer review of "A Multicenter Machine Learning-Based Predictive Model of Acute Toxicity in Prostate Cancer Patients Undergoing Salvage Radiotherapy (ICAROS Study)"

_cancers, 2025, doi:10.3390/cancers17132142_

Round 1
Reviewer 1 Report
Comments and Suggestions for Authors Thank you for your effort to submit the captioned manuscript.The topic is relevant to latest development regarding prostate RT toxicity prediction. However, substantial amount of revision is expected on reporting and experimental technique before the paper can be accpeted by publish. Please revise according to the following suggestions. First, the aim of the work should focus on classification/prediction performance of the model. However, the reported result focus on clinical characteristics of the patient cohort involved in the study, which does not address the aim at all. Please provide all the relevant performance metrics: AUC, precision-recall curve, calibration curve, nonmogram (if applicable), decision curve analysis for all models constructed. LASSO is a machine learning technique, which is NOT generally regarded as AI. According to the method section, the clinical endpoint is solely based on the available data. This is not scientifically sound at all. Please provide a valid clinically relevant reason of choosing acute Grade 2 GU toxicity as the endpoint. In line 116-117, what does it mean by "with almost two increments"? Please refrain from any ambiguity in reporting. Please give the inclusion exclusion criteria in greater detail. E.g. age? previous cancer? history on hormonal treatment unrelated to prostate disease? history of TURP? history of benign prostatic hyperplasia etc. Also, line 128, why 15 node as threshold? please justify with reference. Please state ADT in detail, which drug, category, regimen, etc. Please provide a figure to specify the training and validation split, together with event and non-event ratio in both training and validation cohorts. Regarding Table 1, please also provide details of patient characteristics in each training and validation cohort, including the event and non-event cases in each cohort. After that, please perform statistical analysis as appropriate to confirm any significant differences in the distribution of patient charateristics in each cohort. Further more, please provide comparison and statistical testing for patient characteristics distribution across all centers involved in this study. Please supplement with the treatment protocol of each clinical centre as well. In addition, some key facts are not give, e.g. TNM stage, GS score, risk group, medication, RT fractionation scheme in full. Table 2 is not entirely relevant. Such table should be used for comparing with other studies with the aim to construct and validate models, not investigating a clinical appraoch. Figure 1 and 2 only represents patient characteristics. It does not formulate a model and gives no information in model performance. Please combine it with table 1. For results, please provide correlation matrix between features for input. Also, remove highly correlated features beforehand to avoid overfitting. Please consider to remove clinically irrelevant features before entering LASSO by statistical testing. Please state all the hyperparameters for tuning the LASSO models. Please state the validation cohort is not included in any part of the training. Please clarify if any bootstrapping strategy was employed during training and the reason. Please provide the model signature in the form of a polynomial expression. Please follow strictly TRIPOD guideline or TRIPOD AI+ guideline for reformatting, clarification and addition of information throughout the manuscript to maintain high transparency and reproducibility of results. Please also advice if there are any robustness testing for the LASSO model, such as through noise additon or perturbation of the input features. The direction of discussion is not highly relevant to the model itself, but clinical tolerability of SRT, which does not match with the aim of the study at all. Please re-do the discussion, by comparing the modelling strategy with other published models, elaborating on what research gap this model has filled and why. The recommendation should lie on what can be done to mitigate the toxicities. It is generally agreed that a certain margin is required for treatment efficacy. The role of such toxicity prediction model should lie on early identification of pateints and thus allow timely mitigative measures to tackle the toxicities issues. Comments on the Quality of English Language
Too many grammatical mistakes to be listed. Please maintain a high standard of written English throughout.
Author Response
Comment 1: The topic is relevant to latest development regarding prostate RT toxicity prediction. However, substantial amount of revision is expected on reporting and experimental technique before the paper can be accepted by publish. Please revise according to the following suggestions. First, the aim of the work should focus on classification/prediction performance of the model. However, the reported result focus on clinical characteristics of the patient cohort involved in the study, which does not address the aim at all. Please provide all the relevant performance metrics: AUC, precision-recall curve, calibration curve, nomogram (if applicable), decision curve analysis for all models constructed. LASSO is a machine learning technique, which is NOT generally regarded as AI.
Response 1: Many thanks for your suggestions. Following your recommendations, the manuscript was revised in order to better focus on the classification/prediction performance of the models. We provided all the relevant performance metrics as new tables and figures. Lastly, we agree that LASSO cannot regarded as AI. This regression technique was used for variable selection, especially when a large amount of predictor variables are available, in order to obtain more parsimonious, and hence, more interpretable prediction models.
Comment 2: According to the method section, the clinical endpoint is solely based on the available data. This is not scientifically sound at all. Please provide a valid clinically relevant reason of choosing acute Grade 2 GU toxicity as the endpoint.
Response 2: Thank you for pointing out the need to justify our choice of acute Grade 2 toxicity (both genitourinary and gastrointestinal) as the study endpoint. We have streamlined the explanation in the Study design and endpoints subsection so that it focuses solely on the clinical relevance of Grade 2 events, as requested.
- Original manuscript text that has been modified
“We selected grade 2 acute toxicity as the specific endpoint for this analysis, given that the incidence of grade 3 acute GI and GU toxicity was below 1 %.”
- Re-written (new) text now appearing in the manuscript
“Acute Grade ≥ 2 GU and GI toxicities (RTOG scale) recorded from the start of radiotherapy to 90 days post-treatment were chosen as the clinical endpoints because Grade 2 represents the first toxicity level at which symptoms (e.g., dysuria requiring α-blockers, ≥4 extra bowel movements per day needing loperamide) interfere with daily activities and necessitate medical intervention, thereby impacting quality of life and potentially prompting treatment adaptation.”
- Location of the new text
Section 2.1 “Study design and endpoints,” second paragraph (immediately after the sentence describing the retrospective multicenter design).
Comment 3: In line 116-117, what does it mean by "with almost two increments"? Please refrain from any ambiguity in reporting.
Response 3: Thank you for drawing our attention to the ambiguous wording in lines 116–117. We agree that the phrase “with almost two increments” is unclear and have replaced it with a precise definition of biochemical recurrence.
- Original manuscript text that has been modified
“…3) rise of PSA with almost two increments and a value > 0.2 ng/mL…”
- Re-written (new) text now appearing in the manuscript
“…3) biochemical recurrence defined as at least two consecutive increases in PSA, with the last measurement > 0.2 ng/mL…”
- Location of the new text
Section 2.2 “Inclusion and exclusion criteria,” bullet #3 of the inclusion criteria list.
Comment 4: Please give the inclusion exclusion criteria in greater detail. E.g. age? previous cancer? history on hormonal treatment unrelated to prostate disease? history of TURP? history of benign prostatic hyperplasia etc.
Response 4: Thank you for requesting a more detailed description of our inclusion and exclusion criteria. We have rewritten Section 2.2 in narrative form and clarified that age, previous cancers, hormonal therapy unrelated to prostate disease, prior TURP, and benign prostatic hyperplasia were not used as exclusion criteria.
1.Original text that has been modified
“The following inclusion criteria were used: 1) patients with prostatic adenocarcinoma who underwent previous RP, 2) absence of distant metastases, 3) rise of PSA with almost two increments and a value > 0.2 ng/ml, 4) RT delivered with external beams techniques using photons beams. Also patients with macroscopic recurrent disease in the prostatic bed were included. Exclusion criteria were as follows: 1) previous RT on the pelvic region, 2) contraindication to RT (active inflammatory bowel diseases, pelvic abscesses or fistulas), 3) previous salvage therapy of biochemical recurrence with ADT.
- Re-written (new) text
“Eligible patients were men with histologically confirmed prostatic adenocarcinoma treated by radical prostatectomy who showed no evidence of distant metastases on conventional imaging or PSMA-PET and had biochemical recurrence, defined as at least two consecutive PSA rises with the last value greater than 0.2 ng/mL; all patients received external-beam photon salvage radiotherapy (3D-CRT, IMRT or VMAT), and the presence of macroscopic local recurrence in the prostate bed did not preclude inclusion. Patients were excluded only if they had previously undergone pelvic radiotherapy, had absolute contraindications to pelvic irradiation such as active inflammatory bowel disease, pelvic abscess or fistula, or had already received androgen-deprivation therapy as salvage treatment for biochemical recurrence. No upper age limit was applied, and a history of other malignancies, TURP, benign prostatic hyperplasia, or hormonal therapy unrelated to prostate cancer did not influence eligibility.”
3.Location of the new text
Section 2.2 “Inclusion and exclusion criteria”, entire paragraph replacing the former.
Comment 5: Also, line 128, why 15 node as threshold? please justify with reference.
Response 5: Thank you for asking us to justify the choice of fifteen nodes as the threshold for classifying the extent of pelvic lymph-node dissection (PLND). We have revised Section 2.3 so that the rationale is explicit and we have added the supporting citation to the reference list.
1.Original text that has been modified
“…whether lymphadenectomy was performed, including the extent of lymph-node dissection (< 15 or ≥ 15 nodes).”
2.Re-written (new) text
“…whether lymphadenectomy was performed and, if so, the extent of dissection, classified as limited (< 15 nodes removed) or extended (≥ 15 nodes removed); the cut-off of fifteen nodes was adopted because an extended PLND with a median of fifteen nodes excised is associated with only a moderate increase in surgical morbidity compared with limited dissection and lower complication rates than the super-extended approach that typically removes twenty or more nodes (Seikkula H et al., Front Oncol 2017; 7:280)”
Moreover, we added the following reference to the references list.:
“Seikkula H, Janssen P, Tutolo M, Tosco L, Battaglia A, Moris L, Van den Broeck T, Albersen M, De Meerleer G, Van Poppel H, Everaerts W, Joniau S. Comparison of Functional Outcome after Extended versus Super-Extended Pelvic Lymph Node Dissection during Radical Prostatectomy in High-Risk Localized Prostate Cancer. Front Oncol. 2017;7:280.”
3.Location of the new text
Section 2.3 “Evaluated parameters,” third sentence (description of surgical factors).
Comment 6: Please state ADT in detail, which drug, category, regimen, etc.
Response 6: Thank you for requesting a more detailed specification of the androgen-deprivation therapy (ADT) administered in our cohort. We have modified the paragraph 3.1 to describe the drug categories and regimens used.
1.Original text that has been modified
“This analysis included a total of 454 patients treated across three Italian radiotherapy departments. Patients and treatment characteristics are detailed in Table 1.”
2.Re-written (new) text
“This analysis included a total of 454 patients treated across three Italian radiotherapy departments. Sixty-four per cent of patients received androgen-deprivation therapy (ADT), of whom 47.4 % were treated with a luteinizing hormone-releasing hormone (LH-RH) agonist (triptorelin acetate 11.25 mg every three months or leuprorelin acetate 11.25 mg every three months) or with high-dose bicalutamide (150 mg daily). Patients and treatment characteristics are detailed in Table 1.”
3.Location of the new text
Section 3.1 “Patients characteristics”, first paragraph.
Comment 7: Please provide a figure to specify the training and validation split, together with event and non-event ratio in both training and validation cohorts.
Response 7: Thanks for this suggestion. We added a new figure (Figure 1) to specify the whole workflow of the study, including the training validation split.
Comment 8: Regarding Table 1, please also provide details of patient characteristics in each training and validation cohort, including the event and non-event cases in each cohort.
Response 8: Thanks for this suggestion. Now Table 1 provide the details of patient characteristics in the training and validation cohort. Statistical tests were performed to highlight potential differences between the two cohorts.
Comment 9: After that, please perform statistical analysis as appropriate to confirm any significant differences in the distribution of patient characteristics in each cohort.
Response 9: As reported in the previous comment, statistical analysis was performed to highlight if significant differences are present in the distribution of patient characteristics in each cohort. Statistical comparisons were computed using independent t-test for continuous variables and chi-square test for categorical variable. A p-value of < 0.05 indicates a significant difference.
Comment 10: Furthermore, please provide comparison and statistical testing for patient characteristics distribution across all centers involved in this study.
Response 10: Thanks for this suggestion. Unfortunately, we are not able at the moment to provide a comparison for patients characteristics across the three centers. All three datasets were combined and randomly divided into training and validation sets, with and 70:30 ratio.
Comment 11: Please supplement with the treatment protocol of each clinical center as well.
Response 11: Thank you for asking us to supplement the manuscript with information on the treatment protocols used at each participating center. We have modified the text in the Methods to explain the center-specific variability.
1.Original text that has been modified
“Treatment characteristics in this cohort, including dose, fractionation, prophylactic lymph node irradiation, and combination with ADT, varied according to departmental guidelines and the judgment of the attending physicians.”
2.Re-written (new) text
“Treatment characteristics in this cohort, including dose, fractionation, prophylactic lymph-node irradiation and combination with ADT, varied according to departmental guidelines and the judgment of the attending physicians. Treatment protocols were not uniform across individual participating centers; they evolved over time in line with available technological resources (e.g. transition from 3D-CRT to IMRT/VMAT), the progressive spread of mild hypofractionation in prostate cancer and the personal preferences of the responsible radiation oncologists.”
3.Location of the new text
Section 2.1 “Study design and endpoints”, final sentence of the paragraph describing treatment characteristics (immediately before the sentence on set-up verification).
Comment 12: In addition, some key facts are not give, e.g. TNM stage, GS score, risk group, medication, RT fractionation scheme in full.
Response 12: Thank you for noting the absence of several tumor- and treatment-related variables in the original version of Table 1. We have updated the table to include pathological TNM stage, pre-operative Gleason score, D’Amico risk group, and the complete radiotherapy fractionation scheme. Unfortunately, concomitant medical therapies other than androgen-deprivation therapy were not recorded in our retrospective database and therefore cannot be reported.
1.Original text that has been modified
“Patients and treatment characteristics are detailed in Table 1.”
2.Re-written (new) text
“Patients, tumor, and treatment characteristics, including pathological TNM stage, Gleason score, D’Amico risk classification, and the full radiotherapy fractionation scheme, are detailed in Table 1. Concomitant medical therapies other than androgen-deprivation therapy were not collected in the source databases and are therefore unavailable for analysis.”
3.Location of the new text
Section 3.1 “Patient characteristics,” first paragraph, replacing the original sentence referring to Table 1.
List of data added to table 1:
|
ISUP |
n |
% of total |
|
1 |
77 |
17.0 % |
|
2 |
83 |
18.3 % |
|
3 |
117 |
25.8 % |
|
4 |
86 |
18.9 % |
|
5 |
91 |
20.0 % |
|
pT stage |
n |
% of total |
|
1 |
4 |
0.9 % |
|
2 |
183 |
40.3 % |
|
3 |
261 |
57.5 % |
|
4 |
6 |
1.3 % |
|
pN stage |
n |
% of total |
|
0 |
392 |
86.3 % |
|
1 |
62 |
13.7 % |
|
EAU risk category |
n |
% of total |
|
Very low – low |
11 |
2.4 % |
|
Intermediate |
38 |
8.4 % |
|
High |
263 |
57.9 % |
|
Locally advanced |
142 |
31.3 % |
|
Total dose to the prostatic fossa (Gy) |
< 70 Gy |
264 (58.1%) |
|
|
≥ 70 Gy |
190 (41.9%) |
|
Dose per fraction to the prostatic fossa (Gy) |
≤ 2.0 Gy |
168 (37.0%) |
|
|
2.1 – 2.4 Gy |
122 (26.9%) |
|
|
> 2.4 Gy |
164 (36.1%) |
Comment 13: Table 2 is not entirely relevant. Such table should be used for comparing with other studies with the aim to construct and validate models, not investigating a clinical approach.
Response 13: Thank you for pointing out that the original Table 2 was not essential to the clinical focus of the manuscript. Consequently, we have removed Table 2 and integrated the most pertinent comparative remarks directly into the Discussion section.
1.Original text that has been modified
“Table 2: Overview of selected studies, including the present series, analyzing toxicity in post-prostatectomy radiotherapy.”
2.Re-written (new) text
The table has been deleted. Key points of comparison with previous predictive-model studies are now summarized narratively in the Discussion, as follows: “Nowadays, a number of researches employed ML-based models for the prediction of radiotherapy-related side effects, including acute and late gastrointestinal and genitourinary toxicities in prostate treatments [25-32]. Back in 2011, Pella et al. [26] made a first attempt to apply ML techniques for the prediction of acute toxicity for urinary bladder and rectum, reporting a prediction accuracy of AUC = 0.7 using an artificial neural network derived from a cohort of 321 patients. Ospina et al [27] proposed a random forest normal tissue complication probability model to predict late rectal toxicity following prostate cancer radiation therapy, and to compare its performance to that of classic NTCP models. The age and use of anticoagulants were found to be predictors of rectal bleeding, with an AUC ranging from 0.66 to 0.76, depending on the toxicity endpoint. The model included variables other than the dosimetric ones, and was considered as a strong competitor to classic NTCP models. Jones et al. [30] sought to help identify patients at higher risk of developing rectal injury based on estimated rectal dosimetry compliance prior to planning procedure. Logistic regression, classification and regression tree, and random forest models were compared for their ability to discriminate between class outcomes. Both logistic regression and random forest modeling approaches demonstrated good discriminative ability for predicting class outcomes in the upper dose levels. More recently, also the role of genomics and radiomic features as new variables in machine learning modeling was evaluated to predict radiation-induced rectal toxicities [31,32]. Lee et al. [31] developed a random forest regression method for predicting the risk of GU toxicity by identifying and integrating patterns in genome-wide single nucleotide polymorphisms. Gene ontology analysis highlighted key biological processes, such as neurogenesis and ion transport, from the genes known to be important for urinary tract functions. In the study of Yang et al. [32], 183 patients were included and toxicity scores were prospectively collected after 2 years with grade ≥ 1 proctitis, hemorrhage and GI recorded as the endpoints of interest. The test set AUC values were 0.549, 0.741 and 0.669 for proctitis, hemorrhage and GI toxicity prediction using radiomic combined with dosimetric features.
However, the application of these approaches in clinical routine is still lagging, mainly due to their low interpretability, i.e. they are uninterpretable with respect to the importance of each feature on the output prediction and are generally regarded as "black boxes". This is undesirable because clinicians will lose trust as a result of the prediction's unclear understanding.”
Comment 14: Figure 1 and 2 only represents patient characteristics. It does not formulate a model and gives no information in model performance. Please combine it with table 1.
Response 14: While Table 1 refers to the patient’s characteristics, Figures 1 and 2 do not represent patient characteristics but they are representations of the results as obtained by the decision tree models for both acute GI and GU toxicities. Following your previous suggestion (comments 8 and 9) table 1 was revised in order to consider both training and validation cohorts and statistical comparison between the two groups. Then, the figures were redraw as classification trees in order to better explain the role of the most informative variables. The goal was to obtain interpretable decision tree diagrams, i.e. a type of flowchart that simplifies the decision-making process by breaking down the different paths of action available. These diagrams also showcase the potential outcomes involved with each path of action, helping to quickly visualize the different options and explore the potential consequences of a decision.
Comment 15: For results, please provide correlation matrix between features for input.
Response 15: Many thanks for this suggestion. We added a new figure showing the results of the LASSO regression and the correlation matrix between the selected features for input of the different machine learning models.
Comment 16: Also, remove highly correlated features beforehand to avoid overfitting.
Response 16: As explained in the previous comment, this was done.
Comment 17: Please consider to remove clinically irrelevant features before entering LASSO by statistical testing.
Response 17: Thanks for this suggestion. Before LASSO, we already performed a univariate analysis to remove clinically irrelevant features. This is now explicitly written in the text.
Comment 18: Please state all the hyperparameters for tuning the LASSO models.
Response 18: Many thanks for this suggestion. In lasso regression, the hyperparameter lambda (λ), also known as the L1 penalty, balances the tradeoff between bias and variance in the resulting coefficients. As λ increases, the bias increases, and the variance decreases, leading to a simpler model with fewer parameters. Conversely, as λ decreases, the variance increases, leading to a more complex model with more parameters. We have now reported the lambda values for both the LASSO regressions (both GI and GU toxicities).
Comment 19: Please state the validation cohort is not included in any part of the training.
Response 19: Obviously, the validation cohort was never included in any part of the training process. This is now explicitly written in the text.
Comment 20: Please clarify if any bootstrapping strategy was employed during training and the reason.
Response 20: Thanks for this comment. We employed boostrapping to better validate the models performance. In particular, the resulting classification metrics (AUC, F1, accuracy) were averaged across 100 bootstrap. This is now included in the text (paragraph 2.5.), with the following sentences: “The uncertainty of models performance was quantified by bootstrap analysis, with 95% CI intervals based on 100 bootstrap samples. Specifically, for each resample, a new training set was generated by randomly selecting patients with replacements from the original training dataset. The remaining patients were used as validation cohorts. This process was repeated 100 times to assess the robustness of the models.”
Comment 21: Please provide the model signature in the form of a polynomial expression.
Response 21: As explained in the comment 14, the main goal of the paper was to provide decision trees, that is simple interpretable diagrams showing the different choices and their possible results to help clinicians to make decisions easily. The hierarchical tree structure support decision-making by visualizing outcomes, allowing an easy evaluation and comparison of the "branches" to determine which course of action is best. A model signature in the form of a polynomial expression is not easy to produce from decision trees because no coefficients are provided (as in the case of logistic regression or other types of regression).
Comment 22: Please follow strictly TRIPOD guideline or TRIPOD AI+ guideline for reformatting, clarification and addition of information throughout the manuscript to maintain high transparency and reproducibility of results.
Response 22: Thanks for this suggestion. Our procedures were compliant with the 2021 WHO guidance on ethics and governance of artificial intelligence for health and with the TRIPOD+AI statement. This was now explicitly reported in the text. Two references were added [19,20].
Comment 23: Please also advice if there are any robustness testing for the LASSO model, such as through noise addition or perturbation of the input features.
Response 23: We are aware that model robustness is an important issue, referring to the ability of a machine learning model to maintain its performance and make accurate predictions even when it encounters data points or scenarios that differ from those in the training data. In other words, a robust model should be able to handle variations, noise, outliers, or changes in the input without a significant drop in performance. In particular, as several authors have noted, LASSO may be sensitive to outliers in the data. We did not performed a robustness test for the LASSO model, for example with a noisy data test or a perturbation test of the input features. However, every effort was made to handle and remove the outliers in the pre-modeling data processing to improve the model accuracy, reduce bias and enhance interpretability. In addition, we want to underline that decision trees are very poorly sensitive to outliers, and that these are often chosen to reduce the impact of outliers on the model.
Comment 24: The direction of discussion is not highly relevant to the model itself, but clinical tolerability of SRT, which does not match with the aim of the study at all. Please re-do the discussion, by comparing the modelling strategy with other published models, elaborating on what research gap this model has filled and why. The recommendation should lie on what can be done to mitigate the toxicities. It is generally agreed that a certain margin is required for treatment efficacy. The role of such toxicity prediction model should lie on early identification of patients and thus allow timely mitigative measures to tackle the toxicities issues.
Response 24: Thanks for this suggestion. The paper was deeply revised, focusing on the role or ML models and their performance. The comparison of our ML strategy with other published models was included in the discussion, explaining the potential advantages of our approach. In this context, several references were added [25-32]. The recommendation on what can be done to mitigate the toxicities was deeply discussed. We reinforce the benefits of minimally invasive surgery for optimizing both oncological and functional outcomes, including toxicities. We also recommended to reduce the CTV to PTV margin to <10 mm, particularly in open prostatectomy cases, to minimize acute toxicity, suggesting the intensive use of IGRT or motion management techniques such as rectal balloons. The role of toxicity prediction models in the form of decision trees allows the early identification of patients and timely mitigative measures to tackle the toxicities issues. This was now inserted in the discussion section.
Comment 25: Too many grammatical mistakes to be listed. Please maintain a high standard of written English throughout.
Response 25: Thank you for drawing our attention to the grammatical issues. We have asked our native-English-speaking co-author (M.B.) to carry out a full language edit of the manuscript, and all identified errors have been corrected and are visible in the tracked-changes version.
Reviewer 2 Report
Comments and Suggestions for Authors
This paper demonstrated the interplay between patient characteristics, surgical techniques, and radiotherapy parameters in determining acute toxicity after salvage radiation therapy (SRT). Their findings may support the importance of personalized treatment strategies to minimize acute toxicity.
And the methodology using LASSO and CART is theoretically sound.
However, as mentioned in the text, this study gives little meaning to the current situation where IGRT by CBCT with minimal CTV to PTV margin is now normally used in SRT. I cannot think of how to make use of a study that includes only 10% robot-assisted surgery in the future.
Since authors have valuable data from a large number of SRTs, it would be appropriate to explore utterly novel risk factors or to submit the data to a lower level journal.
Minor comment: 3.3. The adverse event rates for 3D-CRT and VMAT in the paragraph are not shown in Figure 1.
Author Response
Comment 1: This paper demonstrated the interplay between patient characteristics, surgical techniques, and radiotherapy parameters in determining acute toxicity after salvage radiation therapy (SRT). Their findings may support the importance of personalized treatment strategies to minimize acute toxicity. And the methodology using LASSO and CART is theoretically sound.
Response 1: Thank you very much for your encouraging appraisal of our work and your endorsement of the LASSO + CART methodology. We are pleased that the reviewer recognizes the value of our findings for personalized treatment strategies aimed at minimizing acute toxicity during and after salvage radiotherapy.
Comment 2: However, as mentioned in the text, this study gives little meaning to the current situation where IGRT by CBCT with minimal CTV to PTV margin is now normally used in SRT. I cannot think of how to make use of a study that includes only 10% robot-assisted surgery in the future.
Response 2: Thank you for this important comment. We acknowledge that the treatment techniques and surgical approaches represented in our cohort do not fully reflect current standards at many high-volume centers, particularly the widespread use of daily IGRT with CBCT and the predominance of robot-assisted prostatectomy. However, we believe the findings remain relevant for several reasons. First, the data reflect real-world clinical practice across multiple centers and over an extended period, including diverse techniques still in use today. Second, the identification of modifiable radiotherapy-related predictors, such as CTV-to-PTV margin, can still inform decisions in modern IGRT workflows, where the temptation to relax margins must be balanced against organ motion and setup uncertainties. Third, the relatively low rate of robotic surgery in our series, while a limitation, also highlights how surgeon-dependent variability (e.g. anastomotic technique, nerve-sparing) may influence outcomes, and supports the value of including surgical factors in toxicity modelling.
1.Original text that has been modified
No explicit acknowledgement of these limitations or their implications was included in the original text.
2.Re-written (new) text
“Furthermore, the relatively low proportion of patients treated with robotic prostatectomy and the limited use of CBCT-based daily IGRT, which is now common practice. However, the multicenter nature of the cohort reflects real-world heterogeneity, and the identified predictors, particularly CTV-to-PTV margin, remain applicable to modern workflows, where careful balance is needed between margin reduction and the risk of geographic miss.”
3.Location of the new text
Section 4 “Discussion”, paragraph on limitations of the study.
Comment 3: Since authors have valuable data from a large number of SRTs, it would be appropriate to explore utterly novel risk factors or to submit the data to a lower level journal.
Response 3: Thank you very much for your valuable suggestion. Indeed, our database was initially designed approximately 10 years ago; it therefore included extensive patient- surgery-, and radiotherapy-related data but did not encompass other emerging factors potentially associated with toxicity, such as inflammation indices, body-composition parameters (e.g. sarcopenia, sarcopenic obesity), and genetic biomarkers. Recognizing the importance of these novel predictors, we are currently planning a prospective study at our institution specifically designed to investigate the role of inflammation indices and body composition in patients with prostate cancer. This future study aims to comprehensively evaluate the impact of these factors on clinical outcomes as well as acute and late radiotherapy toxicities.
1.Original text that has been modified
No previous text explicitly addressed emerging predictors such as inflammation, body composition, or genetics.
2.Re-written (new) text
"Finally, our database was initially established nearly a decade ago and, although extensive, did not include emerging predictors potentially associated with toxicity, such as inflammatory indices, body-composition metrics (e.g. sarcopenia), or genetic biomarkers. A prospective study is currently being planned at our institution to specifically investigate inflammation indices and body composition in prostate cancer patients, assessing their potential effects on both clinical outcomes and acute and late radiotherapy toxicity."
3.Location of the new text
Section 4 "Discussion", final paragraph (immediately before the Conclusions section).
Comment 4: Minor comment: 3.3. The adverse event rates for 3D-CRT and VMAT in the paragraph are not shown in Figure 1.
Response 4: Thank you very much for pointing out this issue regarding the adverse event rates for 3D-CRT and VMAT techniques mentioned in section 3.3. Please note that Figure 1 has now been replaced with a Classification and Regression Tree (CART) graphical representation, which provides a more complete, precise, and clear visualization of the predictive factors and associated acute toxicity rates, including details about radiotherapy techniques.
Reviewer 3 Report
Comments and Suggestions for Authors
Salvage radiotherapy offers a second chance for cure in patients with prostate cancer and most studies show that toxicity is not a very important issue. The authors describe a large retrospective cohort of patients receiveing salvage radiotherapy.
Some important issues a re missing, e.g., what was interval between surgery and salvage treatment? were there complications during surgery? Did patients have GU/GI symptoms before the salvage treatment?
These are a number of critical issues to deternine the toxicity of salvage radiotherapy.
The fact remains that patients do need a second chance of cure
Author Response
Comment 1: Salvage radiotherapy offers a second chance for cure in patients with prostate cancer and most studies show that toxicity is not a very important issue. The authors describe a large retrospective cohort of patients receiving salvage radiotherapy. Some important issues are missing, e.g., what was interval between surgery and salvage treatment? Were there complications during surgery? Did patients have GU/GI symptoms before the salvage treatment? These are a number of critical issues to determine the toxicity of salvage radiotherapy.
Response 1: Thank you for your positive overall assessment of our study and for emphasizing the clinical relevance of factors such as the interval between surgery and salvage radiotherapy, peri-operative complications, and baseline GU/GI symptoms. Unfortunately, these variables were not captured in the retrospective databases of the three participating centers and are therefore unavailable for analysis. We nonetheless believe the present work remains informative because it integrates a wide range of other patient- surgery- and treatment-related variables (n > 25) and applies machine-learning methods to a large multicenter cohort.
1.Original text that has been modified
The manuscript originally contained no explicit statement regarding the absence of data on surgical-to-SRT interval, surgical complications, or baseline symptoms.
2.Re-written (new) text
“We acknowledge as a limitation that our dataset did not include the interval between prostatectomy and salvage radiotherapy, peri-operative surgical complications, or baseline genitourinary/gastrointestinal symptoms, all of which may influence acute toxicity. Future prospective studies should collect these variables systematically.”
3.Location of the new text
Section 4 “Discussion”, penultimate paragraph, immediately after the sentence beginning “However, some limitations must be acknowledged.”
Comment 2: The fact remains that patients do need a second chance of cure.
Response 2: Thank you for stressing that salvage radiotherapy (SRT) provides many patients with a vital second chance at cure. We completely agree that the possibility of Grade 2 toxicity, while relevant, should never by itself be a reason to deny SRT to men with biochemical recurrence. Rather, our results can help clinicians identify those at higher risk and institute closer monitoring and early supportive measures.
1.Original text that has been modified
“SRT demonstrated excellent tolerability. Surgical technique, CTV-to-PTV margin, and treatment parameters were key predictors of toxicity. These findings emphasize the need for personalized treatment strategies integrating surgical and radiotherapy factors to minimize toxicity and optimize outcomes in prostate cancer patients.”
2.Re-written (new) text
“Importantly, the manageable nature of Grade 2 events means that toxicity risk should not deter offering SRT to suitable patients with biochemical recurrence; instead, our model allows clinicians to identify individuals at higher risk, monitor them more closely during treatment and the early post-treatment period, and initiate supportive therapies (such as α-blockers or loperamide) at the first sign of symptoms.”
3.Location of the new text
Section 4 “Discussion”, 3rd paragraph.
Reviewer 4 Report
Comments and Suggestions for Authors
The manuscript developed machine learning models to help identify key factors for acute toxicity in prostate cancer patients. While the machine learning model usages have been largely discussed in the introduction and methods part, there’s no specific discussion of the model metrics and coefficients in the results section, only comparative results are presented, which may deviate from the topic.
- As mentioned above, my main concern is that no model-related results were presented. Results such as which features were selected from LASSO and what are their coefficients, what’s the AUC for the CART and what’s the feature importance or cut-off learned from the CART should be included.
- The intuition of machine learning algorithms selection is missing. Could the author specify why CART was selected as the prediction model? Why not LASSO, random forest or others? Are there any specific considerations here?
- Details about model implementation will be needed: What’s the specific prediction task? Is it a binary classification (Toxicity vs. No toxicity) or multiclass classification (different categories of toxicity)? What’s the hyperparameter setting for the LASSO?
- Line 152: “The dataset was divided into training and validation sets in a 70:30 ratio, with fivefold cross-validation applied to ensure robust model accuracy.” I didn’t follow here. If it’s 5-fold cross validation, why training and validation sets ratio is 7:3 instead of 4:1?
Author Response
Comment 1: The manuscript developed machine learning models to help identify key factors for acute toxicity in prostate cancer patients. While the machine learning model usages have been largely discussed in the introduction and methods part, there’s no specific discussion of the model metrics and coefficients in the results section, only comparative results are presented, which may deviate from the topic.
Response 1: Thanks for this suggestion. Now the results section was deeply expanded. In particular, we added a revised table (Table 1) reporting the patients and treatment characteristics for the total, the training and the validation cohorts, respectively. The comparison between characteristics in training and validation datasets was performed with appropriate statistical tests. We then added a figure reporting the results of the LASSO analysis and the heatmap of correlation matrix. The predictive performances of the different classifiers in the validation cohort were reported in a new table (Table 2). The ROC curves for the five ML models in the training and vali-dation cohorts for GI and GU acute toxicities are shown in a new figure (Figure 2). Figure 3 now reports the calibration curves for the five ML models on the validation set. Lastly, Figure 4 and 5 show the full visualization of the classification tree for the most informative variables for GI and GU toxicities, respectively.
Comment 2: As mentioned above, my main concern is that no model-related results were presented. Results such as which features were selected from LASSO and what are their coefficients, what’s the AUC for the CART and what’s the feature importance or cut-off learned from the CART should be included.
Response 2: Thanks again. As explained in the response of the previous comment, all the requested metrics have been now included in the manuscript, together with the appropriate tables and figures.
Comment 3: The intuition of machine learning algorithms selection is missing. Could the author specify why CART was selected as the prediction model? Why not LASSO, random forest or others? Are there any specific considerations here?
Response 3: Thanks again for the suggestion. After variable selection, we trained and internally validated five different ML models to predict acute toxicity, including logistic regression (LR), Gaussian Naïve Bayes (GNB), K-means neighbors (KNN), Decision Trees (DT) and Light Gradient-Boosting Machine (LGBM). Decision trees (i.e. CART) and LGBM reported the best performances in terms of discrimination and calibration. In addition to these advantages, as also discussed in the discussion section, the choice of decision trees was also considered because of their hierarchical flowchart-like structure, providing a clear and understandable path of decisions. In other words, DT demonstrated not only to improve the prediction accuracy with respect to other algorithms, but also to allow a clinical applicability by efficiently handling complex datasets. DT methods help prioritize the most relevant predictors and provide a decision-making framework that is easily interpretable in clinical practice. In particular, in this paper, decision trees allowed the early identification of patients at risk and timely mitigative measures to tackle the toxicities issues.
Comment 4: Details about model implementation will be needed: What’s the specific prediction task? Is it a binary classification (Toxicity vs. No toxicity) or multiclass classification (different categories of toxicity)? What’s the hyperparameter setting for the LASSO?
Response 4: Thanks for this suggestion. The prediction task was a binary classification for both GI and GU toxicities. The clinical outcomes were binarized as 0 (<2 toxicity) and 1 (≥2 toxicity). This is now explicitly stated in the manuscript. With regard to hyperparameter for the LASSO regression, the key hyperparameter is the regularization strength, here denoted as 'lambda' (λ). It controls the trade-off between minimizing the residual sum of squares (like in ordinary least squares) and penalizing large coefficients. We used the Sci-Kit Learn package in Python that provides the most common hyperparameter optimization technique, i.e. the grid search. With grid search, the researcher defines a search space of possible λ values within which the hyperparameter will be chosen based on its effective minimization of error. The results of lambda parameter are now reported in the results section.
Comment 5: Line 152: “The dataset was divided into training and validation sets in a 70:30 ratio, with fivefold cross-validation applied to ensure robust model accuracy.” I didn’t follow here. If it’s 5-fold cross validation, why training and validation sets ratio is 7:3 instead of 4:1?
Response 5: Thanks for highlighting this typo. The methods part on the ML modeling was now better rewrote.
Round 2
Reviewer 1 Report
Comments and Suggestions for Authors
Thank you very much for the effort in revising the manuscript extensively. This work provides much insight to those in the field of toxicity prediction, especially for prostate cancer.
Comments on the Quality of English LanguageOccasional slips were seen. English editing will be required before publication.
Author Response
REVIEWER 1 – ROUND 2
Comment 1:
Thank you very much for the effort in revising the manuscript extensively. This work provides much insight to those in the field of toxicity prediction, especially for prostate cancer.
Response 1:
Thank you very much for appreciating our effort in improving the manuscript based on your suggestions.
Comment 2:
Occasional slips were seen. English editing will be required before publication.
Response 2:
Thank you for your suggestion. The following correction has been done in the manuscript:
- Line 38: Changed "emphasizing the need for personalized treatment strategies" to "highlighting the need for personalized treatment strategies."
- Line 41: Adjusted "acute gastrointestinal (GI) and genitourinary (GU) toxicities" to "acute gastrointestinal (GI) and genitourinary (GU) toxicity."
- Line 43: Changed "across three Italian radiotherapy centers" to "from three Italian radiotherapy centers."
- Line 48: Modified "lymphadenectomy extent" to "extent of lymphadenectomy."
- Line 50: Changed "grade ≥4 toxicity" to "Grade ≥4 toxicity."
- Line 57: Inserted a comma after "parameters."
- Line 81: Replaced "acute toxicity is also clinically relevant" with "acute toxicity remains clinically relevant."
- Line 91: Added "the" before "traditional statistical methods."
- Line 100: Changed "acute toxicity in SRT for prostate cancer" to "acute toxicity during SRT for prostate cancer."
- Line 117: Changed "individual participating centers" to "participating centers."
- Lines 144 and 145: Replaced "fifteen nodes excised" with "15 nodes excised."
- Line 169: Corrected "Multicategory variables were processed by one-hot encoding" to "Multi-category variables were processed using one-hot encoding."
- Line 177: Changed "were used for the training and validation" to "were used in the training and validation."
- Line 203: Modified "95% CI intervals" to "95% confidence intervals (CIs)."
- Line 210: Adjust "were estimated for each model" to "estimated for each model."
- Line 227: "Sixty-four per cent" corrected to "Sixty-four percent."
- Line 258: Replaced "the number of variables associated with both acute toxicities was reduced" with "the number of variables associated with acute toxicities was reduced."
- Line 278: Changed "to predict acute toxicity in postoperative prostate cancer" to "to predict acute toxicity following postoperative radiotherapy for prostate cancer."
- Line 305: Replaced "illustrates the initial GI toxicity distribution" with "illustrates the distribution of initial GI toxicity."
- Line 353: Corrected "among the different ML classifiers" to "among the ML classifiers."
- Line 498: Changed "Our database was initially established nearly a decade ago and, although extensive, did not include" to "Our database, although extensive, was initially established nearly a decade ago and did not include."
- Line 470: Replaced "Decision trees ML-based models demonstrated to be able" with "Decision tree-based ML models demonstrated the ability."
Reviewer 2 Report
Comments and Suggestions for Authors
The explanatory text for Figure 1 must be included.
Author Response
REVIEWER 2 – ROUND 2
Comment 1:
The explanatory text for Figure 1 must be included.
Response 1:
Many thanks for this suggestion. We have included an explanatory text for the Figure 1 into the manuscript.